# Incremental Ant-Miner Classifier for Online Big Data Analytics

**DOI:** 10.3390/s22062223

**Published:** 2022-03-13

**Authors:** Amal Al-Dawsari, Isra Al-Turaiki, Heba Kurdi

**Affiliations:** 1Computer Science Department, College of Computer and Information Sciences, King Saud University, Riyadh 11451, Saudi Arabia; 437203624@student.ksu.edu.sa; 2Information Technology Department, College of Computer and Information Sciences, King Saud University, Riyadh 11451, Saudi Arabia; ialturaiki@ksu.edu.sa; 3Mechanical Engineering Department, Massachusetts Institute of Technology (MIT), Cambridge, MA 02142-1308, USA

**Keywords:** machine learning, association rule mining, ant colony optimization, incremental classifier, big data analytics, IoT

## Abstract

Internet of Things (IoT) environments produce large amounts of data that are challenging to analyze. The most challenging aspect is reducing the quantity of consumed resources and time required to retrain a machine learning model as new data records arrive. Therefore, for big data analytics in IoT environments where datasets are highly dynamic, evolving over time, it is highly advised to adopt an online (also called incremental) machine learning model that can analyze incoming data instantaneously, rather than an offline model (also called static), that should be retrained on the entire dataset as new records arrive. The main contribution of this paper is to introduce the Incremental Ant-Miner (IAM), a machine learning algorithm for online prediction based on one of the most well-established machine learning algorithms, Ant-Miner. IAM classifier tackles the challenge of reducing the time and space overheads associated with the classic offline classifiers, when used for online prediction. IAM can be exploited in managing dynamic environments to ensure timely and space-efficient prediction, achieving high accuracy, precision, recall, and F-measure scores. To show its effectiveness, the proposed IAM was run on six different datasets from different domains, namely horse colic, credit cards, flags, ionosphere, and two breast cancer datasets. The performance of the proposed model was compared to ten state-of-the-art classifiers: naive Bayes, logistic regression, multilayer perceptron, support vector machine, K*, adaptive boosting (AdaBoost), bagging, Projective Adaptive Resonance Theory (PART), decision tree (C4.5), and random forest. The experimental results illustrate the superiority of IAM as it outperformed all the benchmarks in nearly all performance measures. Additionally, IAM only needs to be rerun on the new data increment rather than the entire big dataset on the arrival of new data records, which makes IAM better in time- and resource-saving. These results demonstrate the strong potential and efficiency of the IAM classifier for big data analytics in various areas.

## 1. Introduction

An online (also called incremental) machine learning model is one of the most effective approaches in machine learning [1,2], where the model is continuously adapted based on arriving data [3,4,5,6,7]. It can be used for various objectives such as clustering, dimensionality reduction, feature selection, reinforcement learning, mining, inference, and data representation [5]. Unlike traditional off-line (also called static) machine learning, and emerging distributed and federated learning models [8], a complete set of training examples does not have to be available before starting the training process of an incremental learning model [9]. Hence, online learning is becoming increasingly essential in the big data analytics domain where datasets evolve over time, such as precision agriculture [10,11], flood prediction [12,13], and business activities [6,14,15].

Due to their success, a growing body of literature has been proposed for developing incremental machine learning models. The complex process of incremental learning usually makes it difficult to find an efficient solution. Therefore, soft computing techniques, such as ant colony optimization (ACO), genetic algorithms (GA) and evolutionary algorithms (EA), are increasingly gaining attention in this field. Soft computing, in contrast to hard (conventional) computing, refers to approximate solutions based on artificial intelligence (AI) that are tolerant of imprecision, uncertainty and partial truth. It provides cost-effective solutions to complex real-life problems in different fields, e.g., [16,17,18].

However, none of the proposed soft computing approaches have considered the Ant-Miner algorithm [19], which is known for its effectiveness in approaching challenging machine learning problems and association rule mining [12,19,20,21,22,23,24,25]. Ant-Miner, which is based on the ant colony optimization algorithm (ACO) [26], is usually exploited in learning classification rules [14,23,27]. In Ant-Miner, each rule has three phases: rule construction, rule pruning, and pheromone updating. In the rule construction phase, the probability of adding a term to the current rule is based on a heuristic function and the pheromone associated with the term. The value of the heuristic function corresponds to the quality of the term. A rule is then pruned in the pruning phase by repeatedly removing irrelevant terms that do not improve the quality of the rule. Finally, in the pheromone updating phase, the pheromone quantities associated with terms appearing in the discovered rule are increased in proportion to the quality of the rule. Additionally, the pheromone associated with terms that do not appear in the rule is decreased. The process of rule building repeats until reaching a user-specified number of rules, or the current constructed rule is the same as the previously constructed rules [11,20].

In this paper, we propose an incremental rule-based machine learning algorithm, Incremental Ant-Miner (IAM), for online prediction, based on Ant-Miner. The system starts with feature extraction and initial classification rules generation from the existing training dataset. Once a new increment to the dataset arrives, the IAM classifier runs only on the new increment to enhance the learning process and optimize the classification rules. IAM was applied to binary and multi-class classification datasets, such as horse colic, breast cancer, credit card, flags, and ionosphere, to evaluate its performance in terms of accuracy, precision, recall, and F-measure. When benchmarking IAM performance against ten rival classifiers, naive Bayes, logistic regression, multilayer perceptron, support vector machine, K*, adaptive boosting (AdaBoost), Bagging, Projective Adaptive Resonance Theory (PART), decision tree (C4.5), and random forest, the results revealed the superiority of IAM in almost all performance measures.

The rest of this paper is structured as follows: Section 2 reviews the related works on applying Ant-Miner-based classifiers and incremental machine learning models in different application areas. The proposed system design and evaluation methodology are presented in Section 3 and Section 4 respectively. Section 5 illustrates and discusses the experimental results. Finally, the conclusions are presented in Section 6.

## 2. Literature Review

In this section we divide the related work into two sets: the first includes the literature that introduced machine learning models based on Ant-Miner while the second includes the works which propose incremental models.

### 2.1. Ant-Miner Models

Parpinelli et al. [19] proposed the Ant-Miner algorithm for data mining. They used six public-domain datasets available in the University of California at Irvine (UCI) repository. The performance of the algorithm was evaluated according to the predictive accuracy. The accuracy was in the range from 59.67% to 96.04%. The developers compared the performances of the proposed Ant-Miner and the learning algorithm for rule induction (CN2) algorithm, on the six datasets. According to the predictive accuracy, the result showed that Ant-Miner is competitive with the CN2 algorithm. Ant-Miner achieved better results in four datasets, whereas CN2 achieved better accuracy in one dataset, and both algorithms obtained similar predictive accuracy in the remaining dataset. Additionally, the Ant-Miner uncovered simpler (smaller) rules than those uncovered by CN2.

Liu et al. [28] used the ant colony optimization (ACO) algorithm for classification rule discovery. They presented a modified version of the Ant-Miner. The presented ACO algorithm used different pheromone levels, updating strategy, and state transition rules. It was applied to two datasets: the Wisconsin breast cancer database and the Tic Tac Toe endgame database, both obtained from the UCI repository. They evaluated the performances of the proposed method using 10-fold cross-validation. The result was compared with the original Ant-Miner. The proposed version uncovered more rules than the original Ant-Miner and achieved greater accuracy values in both the datasets: 94.32% for the breast cancer dataset, and 76.58% for the Tic Tac Toe endgame dataset.

Martens et al. [14] developed a system for credit card prediction based on the Ant-Miner algorithm. The experiment was conducted using different types of datasets: a German credit card dataset from the UCI repository, small- and medium-sized enterprises (SME) dataset, and a banks dataset from the Bankscope database. The accuracy was calculated using the Ant-Miner algorithm based on the three datasets and compared with other classifiers: SVM, C4.5, and majority vote. The accuracy results of the Ant-Miner ranged from 71.9% to 86.2%. The result showed that Ant-Miner and C4.5 had an average accuracy of 80.8%. However, in terms of the number of rules, Ant-Miner performed better.

Nimmy Cleetus et al. [29] used the Ant-Miner algorithm to obtain rules for analyzing different forms of network attacks. They used a dataset with five types of attacks as class values. They used Weka to generate rules for the attacks. By measuring its classification accuracy, the rules created by Weka were not optimal. Therefore, they developed a rule synthesis parser to eliminate the redundant rules, which used the Ant-Miner rule-based classification algorithm. The accuracy results ranged from 90.82% to 100%. Thus, the Ant-Miner algorithm obtained optimal results in this study and had better accuracy compared with the rules that were not synthesized.

Lai et al. [13] proposed a study that used the Ant-Miner algorithm to analyze and zone a flood risk at grid scale. The study was applied in the Dongjiang River Basin in Southern China. The dataset was obtained from previous research in the same region. The researchers applied the Ant-Miner algorithm to two versions of the dataset with or without the socio-economic indices. The proposed model was developed using the MYRA project, which is an Ant-Miner program that was developed using Java languages. The performance accuracy of the proposed Ant-Miner algorithm was compared with the decision tree method in both cases. The Ant-Miner algorithm achieved accuracy of 92.5% and 94.9% in the two cases, with or without the socio-economic indices, respectively. Thus, it was more accurate than the decision tree method.

Durgadevi et al. [24] presented a study that used the Ant-Miner algorithm to extract classification rules for medical distress prediction. They used three benchmark datasets: a heart disease dataset from Cleveland, a diabetes dataset from Pima, India, and a breast cancer dataset from Wisconsin, all from the UCI repository. Preprocessing and feature evaluation methods were applied to the three datasets. Mean selection, half selection, and threshold selection were the three feature evaluation methods applied to reduce the number of features in the datasets. The performance of the proposed Ant-Miner algorithm was evaluated by measuring the accuracy values for the three datasets with different feature evaluation methods. The accuracy of the Ant-Miner was compared with other classifiers, RPF (reverse-path forwarding) network, CN2, AdaBoost, and bagging. The accuracy results of Ant-Miner ranged from 96.57% to 99.85%. The results showed that the Ant-Miner algorithm achieved the best accuracy result among other algorithms in eight cases out of 12.

Sabri et al. [12] used the Ant-Miner algorithm to develop a classification prediction model to predict flooding. The rainfall dataset used to develop the model was obtained from three rainfall gauging stations in Perlis, Malaysia. The dataset was discretized and clustered in the preprocessing step. Discretization was performed using a symbolic aggregate approximation (SAA) technique that was used to convert time-series data to discrete data. Then, the data produced by SAA were clustered using a k-mean algorithm. The classification prediction model was developed using the Ant-Miner algorithm with different values for the number of ants. The lowest accuracy was about 80.60% for 25 ants. The highest accuracy result was 84.29% with 30 ants, which was higher compared to the decision tree (C4.5) accuracy result of 77.27% for the same dataset.

Ramalingam et al. [15] developed an Ant-Miner algorithm-based model for accurately predicting stock prices. The Ant-Miner algorithm was applied to a Dow Jones Index dataset and three other finance datasets from Yahoo. The performance of the proposed Ant-Miner was evaluated using accuracy, F-measure, area under curve (AUC), discriminant power, and Youden’s index. A study was presented to compare the performance of the Ant-Miner on the Dow Jones Index dataset for stock price prediction with other classifiers, namely naïve Bayes, a traditional machine learning (ML)-based classifier, radial basis function kernel (RBF), C4.5, random forest (RF), classification and regression tree (CART), and Olex-GA, a GA-based approach for the induction of rule-based text classifiers. The Ant-Miner algorithm achieved an accuracy of 96%, which was the best result among the classifiers. Moreover, the Ant-Miner was applied to Yahoo finance datasets and achieved accuracies ranging from 56.16% to 84.61%.

To summarize, based on these studies, we can conclude that the Ant-Miner has been used in prediction tasks for constructing offline classification models. In various applications, the Ant-Miner algorithm has achieved good performance and has even outperformed some other data mining techniques. However, based on the above, exploiting the Ant-Miner power in constructing online incremental machine learning models has not been explored yet.

### 2.2. Incremental Models

Zhang and Zhao [30] proposed an online fault prediction model for a nonlinear system. The model was developed by combining sliding autoregressive moving average (ARMA) modeling with online least squares support vector regression (LS-SVR) compensation. It could accurately predict the fault of a nonlinear system, and the result demonstrated the efficiency of the proposed online prediction model.

Gao et al. [31] proposed an incremental model to predict disk failure using the density metric of edge samples. Many features are used by the model that increase the system complexity. Therefore, the researchers identified these features through an incremental structure. The proposed incremental prediction model was evaluated using recall rate and public disk datasets. It outperformed several other algorithms, such as Isolation using Nearest Neighbour Ensemble (iNNE), Isolation forest (iForest), and Local outlier factor (LOF).

Tantisripreecha and Nuanwan [6] developed an online learning model for stock movement prediction based on Linear Discriminant Analysis (LDA). The experiments were applied to National Association of Securities Dealers Automated Quotations stocks (APPLE, FACEBOOK, GOOGLE, and AMAZON). The results were compared with the artificial neural network (ANN), K-Nearest Neighbour (KNN), and decision tree in both batch and online learning approaches and demonstrated that the proposed LDA online model reaches the highest performance among other algorithms in both batch learning and online learning.

Jiang et al. [32] proposed an online prediction model based on matrix factorization to predict retweeting behavior. The traditional models cannot adapt to the increase in messages, which may decrease the prediction accuracy, so the online matrix factorization model was developed. The proposed model was evaluated according to precision, recall, F-measure, and accuracy. The results outperform baselines with higher accuracy and shorter running time.

Bin et al. [2] proposed a new incremental learning SVM algorithm. The proposed method was based on the path following technique in the framework of Difference of Convex (DC) programming. The experiment was conducted on a variety of benchmark datasets, and the results proved the IL-SVM is faster than existing other batch and incremental learning algorithms.

Rojas et al. [7] proposed a performance comparison of traditional and incremental learning algorithms for the consumption behavior of Over the Top (OTT) applications. The results demonstrated that incremental learning is a suitable approach for the changes that the users make in the OTT over time. Additionally, the result analysis revealed that the model combining Ozal bagging and the KNN algorithm was the best classifier using the incremental approach.

Zou and Jun [33] proposed an online early warning prediction system for web server failure based on running time. They obtained the key nodes on the server running path and analyzed the running status data, then they used a combined Long Short-Term Memory-Support vector machine (LSTM-SVM) algorithm to predict the occurrence of failure. The resulting accuracy of the early crash warning was high compared with existing methods.

Tan et al. [34] proposed an online prediction model for video popularity in online videos services (OVSs). The model was built through a video age-sensitive function based on the relation between the average watched percentage and the future views of videos. The performance results demonstrated the effectiveness of the proposed model through a series of experiments.

Lv et al. [35] presented an online prediction model to predict ladle furnace temperatures based on an extreme learning machine (ELM). The ELM model was first constructed, then the online sequential learning was adopted to correct the ELM-based prediction model. The experimental results revealed that the proposed ELM prediction model achieved high accuracy, and the online sequential learning is extremely fast, making it suitable for practical application.

The reviewed literature demonstrates the effectiveness of the incremental learning model for online prediction objectives in various areas. However, to the best of our knowledge, none of the previous work utilized Ant-Miner for online prediction, although it has the potential to result in higher prediction accuracy and a smaller rule list. Therefore, this study aimed to bridge this gap.

## 3. Proposed IAM Classifier Design

Machine learning algorithms, whether offline or online, need to be rerun upon the arrival of a new dataset increment. The difference between the two approaches is that offline classifiers should always be rerun on the entire dataset (the original dataset and the new increment). On the other hand, online classifiers need only to be rerun on the newly arrived increment. Accordingly, they are more time and space efficient for this particular type of dataset. The proposed online classifier, IAM, is not different in this respect. It simply utilizes a well-established rule mining algorithm for online learning, Ant-Miner, which is usually used for offline learning, by enhancing its learning ability each time a new increment to the dataset arrives. In a similar approach to [11], we divided each dataset into three increments to mimic the dynamic arrival of data.

As shown in Figure 1, the process starts by training Ant-Miner on an existing dataset, DS1. First, feature evaluation methods are applied, including correlation and information gain; then, initial classification rules are generated. As a new increment to the dataset arrives, DS_2_, the incremental classifier, IAM, analyzes the new dataset and updates the previous knowledge base dynamically. On arrival of a new data increment, IAM runs only on the new increment to enhance the learning process and optimize the classification rules. Then, IAM is evaluated on a test dataset, DS_3_.

### 3.1. Initial Classifier Generation

The initial set of classification rules is generated by running Ant-Miner on an existing dataset, DS1. Every classification rule is comprised of two parts: antecedent and consequence. The classification rules will have the following format:

IF   antecedent   THEN   consequence

IF <term1> AND <term2> AND …… THEN <class>

Antecedent represents the conditional features and their values, whereas consequence represents the decision feature with the corresponding decision value or class label. Each term in the rule antecedent consists of a feature, operator, and value. The same applies to the rule consequent, which has a class feature, operator, and class label or value. The value belongs to the domain of the feature, whereas the operator element is a relational feature [11,19,36,37]. The current version of Ant-Miner deals with both categorical and continuous features so that the operator element in the triple is always “=” since the continuous (real-valued) features have been discretized by the model before use by the algorithm.

The goal of the Ant-Miner algorithm is to extract classification rules from a dataset by providing a set of rules for each class of objects separately.

#### 3.1.1. Pheromone Initialization

The initial pheromone values are equally distributed. These values are inversely proportional to the number of all feature values [19,27,29,37,38]. It is defined by:(1)$$\tau_{ij}\left(t=0\right)=\frac{1}{\sum_{t=1}^{a}b_{i}}$$
where:

*a* is the total number of features.

*b_i_* is the number of possible values for feature *i*.

#### 3.1.2. Rule Creation

The rule of the Ant-Miner algorithm has two parts: the antecedent, which has the conditional features, and the consequence, which has the predicted class. Initially, the list of discovered rules is empty. The ant starts with an empty rule that does not have a term in the antecedent part, and the training set contains all the training cases in the dataset. The current partial path followed by the ant is represented by a current partial rule created by that ant and the term chosen by the ant corresponding to the direction that extends to that path. The ant starts adding one term at a time to its current partial rule. The quantity of pheromones and the problem-dependent heuristic function associated with each term determine whether that term should be added or not to the current partial rule of the ant. Let us assume a rule conditional part such as *term_ij_* is *A_i_* = *V_ij_*, where *A_i_* is the *i*th attribute and *V_ij_* is its *j*th value. The probability that *term_ij_* will be included in the current partial rule is displayed by:(2)$$P_{ij}\;\left(t\right)=\frac{\tau_{ij}\left(t\right)\times N_{ij}}{\sum_{i=1}^{a}\sum_{j=1}^{b_{i}}\tau_{ij}\left(t\right)\times N_{ij},\;\forall \;i\in I}$$
where:

*N_ij_* is a problem dependent heuristic value for *term_ij_*.

τ*_ij_* is the quantity of pheromone associated with *term_ij_* at iteration *t*.

The ant continues adding one term at a time to its current partial rule until using all features or meeting the user-specified threshold (minimum cases by a rule) [19,27,29,37,38].

#### 3.1.3. Heuristic Function

In the Ant-Miner algorithm, the heuristic value is an information-theoretic value measured for analyzing the quality of a rule, which means measuring the excellence of the term to be added to the rule in terms of the ability of the term to improve the predictive accuracy of that rule. The value of the heuristic function is based on a measure of the quantity of information or entropy associated with the term. The value of the heuristic function is normalized in Equation (3) to facilitate its use in measuring the entropy [19,27,37,38]. The proposed normalized information-theoretic heuristic function and the entropy is given by the following equations:(3)$$N_{ij}=\frac{log_{2}\left(k\right)-InfoT_{ij}}{\sum_{i}^{a}\sum_{j}^{b_{i}}log_{2}\left(k\right)-InfoT_{ij}}$$
(4)$$InfoT_{ij}=\left[\frac{freq\;T_{ij}^{w}}{\left|T_{ij}\right|}\right]\times log_{2\;}\left[\frac{freq\;T_{ij}^{w}}{\left|T_{ij}\right|}\right]$$
where:

*k* is the number of classes,

|*T_ij_*| is the total number of cases in partition *T_ij_* (the partition containing cases where feature *A_i_* has value *V_ij_*),

*freq T_ij_ w* is the number of cases in partition *T_ij_* with class *w*.

The higher the value of *InfoT_ij_*, the more uniformly distributed the classes are, and so, the lower the probability that the current ant will prefer *term_ij_* and add it to its current partial rule.

#### 3.1.4. Rule Pruning

Rule pruning is a common technique usually used in machine learning. It is used to remove one term at a time from the rule if that term has been unduly added in the rule and removing it will improve the quality of the rule. The quality of the rule is measured using the following equation:(5)$$Q=\frac{TP}{TP+FN}\times \frac{TN}{FP+TN}$$
where:

*TP* is the number of cases covered by the rule and has the same class that is predicted by the rule,

*FP* is the number of cases covered by the rule and having a class that was not predicted by the rule,

*FN* is the number of cases that are not covered by the rule while having the class that is predicted by the rule,

*TN* is the number of cases not covered by the rule and having a different class from the class predicted by the rule.

Once the rule is constructed by the ant, it is pruned to remove irrelevant terms from the rule antecedent. Then, the consequence of the rule is chosen to be the most frequent class value among the set of training examples covered by the rule. The rule pruning process increases the predictive accuracy of the rule, which helps in avoiding overfitting problems to the training data. Additionally, rule pruning improves the simplicity of the rule, as the shorter rule is easier to be understood by the user than the longer one. The process of rule pruning is repeated until one of the following criteria is met:The rule has just one term, orThere is no term whose removal will improve the quality of the rule.

#### 3.1.5. Pheromone-Changing Rule

In the rule discovery context, the pheromone-changing rule means updating the probability of the *term_ij_* to be chosen by another ant in the future, whether by increasing or decreasing that probability. After each ant constructs its rule and that rule is pruned, the quantity of pheromones in all terms will be updated as follows:(6)$$T_{ij}\left(t+1\right)=T_{ij}\left(t\right)+T_{ij}\left(t\right)\times Q\;\;\;\;\;\;\forall \;i,j\;\in R$$
where:

*R* is the set of terms that occur in the rule constructed by an ant at iteration,

*Q* is the quality of the rule.

For simulating the phenomenon of evaporation in the ACO system, the quantity of pheromones associated with each *term_ij_* that do not take place in the constructed rule will be decreased. The pheromone reduction of an unused term is performed by dividing the value of each *T_ij_* by the summation of all *T_ij_*.

### 3.2. Dynamic Classifier Generation

In this step, the classifier is trained whenever a new dataset DS_2_ arrives. Initially, DS_2_ generates a rule set by following the same steps as in the initial classifier generation. Then, the new rule set is upgraded using the previously extracted rule set of the initial classifier to develop the dynamic classifier. In this step, the system computes the similarity between each rule in the new ruleset with each rule in the old ruleset if they have the same class value.

**Definition** **1.**
*The similarity value of new rule j with the existing ruleset R_old_ is reflected in Equation (7)**.***


(7)$$S=\frac{1}{R_{new}^{j}}\;max\left(R_{old}^{i}\cap R_{new}^{j}\right)$$
where:

1 ≤ *t* < *n*, for all *j* = 1,2,3, …… *m*

$R_{new}^{j}$ is the rule number j in the new rule list.

$R_{old}^{i}$ is the rule number i in the old rule list.

$max\left(R_{old}^{i}\cap^{\;}R_{new}^{j}\right)$ is the maximum intersection terms between $R_{new}^{j}$ and each rule in the old rule list.

If the similarity between the new rule and the old rule is 1, and the new rule has the same length as the old rule, then the new rule is added to the dynamic rule set, and its quality is upgraded using the fitness function, based on Definition 2. Otherwise, if the similarity between the new rule and the old rule is 1, and the old rule has more terms in the antecedent than the new rule, the old rule is added to the dynamic ruleset. Subsequently, if the similarity between the new rule and the old rule is zero or greater than 0 and less than 1, the new rule is added to the dynamic ruleset without changing the quality.

**Definition** **2.**
*The fitness function is calculated in Equation (8) by taking the weighted sum of the quality of the rule given in Equation (5) and the similarity measure value computed in Equation (7)**.***


(8)$$Fitness=w\times Q_{Rnew}+\left(1-w\right)S$$
where:

*w* is a weight factor set experimentally for each dataset.

*Q_Rnew_* is the quality of the rule in the new list.

Table 1 lists the main parameters of Ant-Miner and IAM and their assigned values. We based the pheromone value on the number of attributes in the dataset and the number of values for each attribute, whereas we used the entropy associated with the term as the base for the heuristic value and the heuristic as the base for the probability value of each term. The pheromone values associated with the term and the quality value of a rule were calculated using *TP*, *FP*, *TN*, and *FN*. The decision to change the pheromone was made according to the values of the pheromone on the pervious iteration and the quality of the rule.

## 4. Methodology

### 4.1. Dataset Selection

The proposed IAM classifier was applied to six different datasets available in the University of California at Irvine (UCI) repository [39]. For simplicity, the selected datasets are relatively small in size due to the limited computational capability of the hardware used to run the models. The main objective is to prove the concept rather than comprehensively evaluate the proposed model. To ensure generality, datasets were selected from different domains, namely horse colic, credit cards, flags, ionosphere, and two breast cancer datasets. The dataset’s descriptions are provided in the corresponding sub-sections of Section 5. The main features of each dataset are summarized in Table 2.

The proposed model deals with discrete-valued datasets and continuous datasets, as datasets are discretized before using the algorithm. The datasets were treated as incremental datasets. Thus, each dataset was divided randomly into three parts: 40% for DS1 as an old dataset, 30% for DS_2_ as an increment to the dataset, and 30% for DS_3_ as a test dataset.

### 4.2. Dataset Preparation

For developing the incremental prediction model, we need each dataset to be partitioned into at least three parts to mimic an evolving dataset:The first part is called the old dataset, denoted by DS_1_. It represents the initial existing dataset and is used to generate the initial rules using the Ant-Miner algorithm.The second part, denoted by DS_2_, mimics a new increment to the dataset. This dataset is used to generate the new rules using the Ant-Miner algorithm. These new rules are used along with the initial rules to find out the final incremental rule list and update the rule quality before adding it to the incremental rule list.The third part, denoted by DS_3_, is the test dataset. It is used to evaluate the proposed system based on the incremental rule list.

### 4.3. Benchmark Algorithms

To evaluate IAM performance, we used ten state-of-the-art machine learning algorithms, in a similar approach to [11]: naïve Bayes [40], K* [41], Ada-Boost, bagging [40], PART [42], decision tree (C4.5), multilayer perceptron (MLP), SVM, random forest, and logistic regression [40].

### 4.4. Performance Measures

We used four performance measures to evaluate the proposed IAM and the benchmark algorithms: accuracy, precision, recall, and F-measure. The evaluation measures were computed based on the average of the classes. Those are denoted by Formulas (9)–(12), respectively.
(9)$$\mathrm{Accuracy}=\frac{TP+TN}{TP+FP+TN+FN}$$
(10)$$\mathrm{Precision}=\frac{TP}{TP+FP}$$
(11)$$\mathrm{Recall}=\frac{TP}{TP+FN}$$
(12)$$F_{\mathrm{measure}}=\frac{2}{\left(\frac{1}{\mathrm{precision}}+\frac{1}{\mathrm{recall}}\right)}$$
where:

*TP* (true positives) is the total number of samples that are correctly classified as “infested”,

*FP* (false positive) represents the total number of samples that are incorrectly classified as “infested”,

*FN* (false negatives) represents the total number of samples that are incorrectly classified as “healthy”, and

*TN* (true negative) represents the total number of samples that are correctly classified as “healthy”.

## 5. Results and Discussion

All experiments were run on the same hardware, i.e., a laptop with the following specifications: MacBook Pro 9.2; Operating System: OS Mojave 10.14.2; Processor: Intel Core 17; Speed: 2.9 GHz; Memory: 8 GB 1600 MHz DDR3. Java was used for the implementation of the proposed IAM, and Weka tools [43] were used for benchmarking IAM with the state-of-the-art machine learning algorithms and dataset pre-processing.

The proposed IAM model was applied to six different datasets: breast cancer (Ljubljana), horse colic, credit card, flags, ionosphere, and breast cancer (Wisconsin) datasets. This section presents the results of applying the proposed IAM model to these datasets and the benchmark algorithms.

### 5.1. The Results of the Breast Cancer (Ljubljana) Dataset

The breast cancer (Ljubljana) dataset has nine features and 286 records that are classified into two classes. We applied the IAM to this dataset by trying different splitting parameters to obtain the best split that provides the highest performance result, as presented in Table 3. The highest values are bolded for each performance measure.

The above results display the highest performance values obtained by the IAM algorithm with 78.18% for accuracy, 89.47% for precision, and 85% for F-measure. Conversely, the highest accuracy of the benchmark algorithms was 75.52%, whereas the highest recall was 96% achieved by C4.5, the highest F-measure was 84.60%, and the highest precision value was 77.8%. These results show that the IAM algorithms outperformed all the state-of-the-art algorithms when applied to the breast cancer (Ljubljana) dataset regarding the accuracy, precision, and F-measure.

### 5.2. The Results of the Breast Cancer (Wisconsin) Dataset

The breast cancer (Wisconsin) dataset has 10 features and 699 records classified into two classes. We applied the IAM to this dataset by trying different splitting parameters to obtain the best split that provided the highest performance result, as presented in Table 4. The highest values are bolded for each performance measure.

The above results reveal that the highest performance values obtained from applying IAM classifier on the breast cancer (Wisconsin) dataset were 98.65% for accuracy, 94.4% for precision, 100% for recall, and 97.1% for F-measure. Conversely, the highest accuracy of the benchmark classifiers was 96.85% achieved by random forest, while the highest precision was 94.7%, achieved by SVM, differing slightly from IAM. The highest recall value was 97.5%, achieved by naïve Bayes, and the highest F-measure was 95.5%, also achieved by random forest. These results demonstrate that the IAM algorithms outperformed all the state-of-the-art algorithms regarding accuracy, precision, and F-measure.

### 5.3. The Results of the Horse Colic Dataset

The horse colic dataset has 22 features and 368 records that are classified into two classes. We applied the IAM on this dataset using different splitting parameters to get the best performance result. Moreover, the benchmark algorithms were applied using 10 cross-validation folds. The results are presented in Table 5. The highest values are bolded for each performance measure.

The highest performance values obtained from applying the IAM algorithm on the horse colic dataset were 86.11% for accuracy, 95.24% for precision, 83.33% for recall, and 88.89% for F-measure. However, the performance accuracy of the benchmark algorithms ranged from 76.63% to 86.41%, whereas the highest values for the other measures were 87% for precision, 93.1% for recall, and 89.6% for F-measure. The above results demonstrate that the IAM, bagging, and random forest classifiers outperformed all the other state-of-the-art classifiers.

### 5.4. The Results of the Credit Card Dataset

The credit card dataset has 15 features and 690 records that are classified into two classes. We applied the IAM to this dataset using different splitting parameters to obtain the best performance result. Additionally, the benchmark algorithms were applied using 10-fold cross-validation. The results are presented in Table 6. The highest values are bolded for each performance measure.

The above results reveal that the highest performance values obtained from applying IAM classifier on the credit card dataset were 92.75% for accuracy, 93.75% for precision, 86.54% for recall, and 90% for F-measure. Conversely, the highest accuracy of the benchmark classifiers was 86.96% achieved by random forest, whereas the highest precision was 85.9%, achieved by naïve Bayes. The highest recall value was 92.2% achieved by SVM, and the highest F-measure was 85.2%, also achieved by random forest. These results show that the IAM algorithms outperformed all the state-of-the-art algorithms regarding accuracy, precision, and F-measure.

### 5.5. The Results of the Flags Dataset

The flags dataset has 30 features and 194 records that are classified into eight classes. We applied the IAM to this dataset using different splitting parameters to obtain the best performance result. Additionally, we applied the benchmark algorithms using 10-fold cross-validation. The results are presented in Table 7. The highest values are bolded for each performance measure.

Table 5 shows that IAM outperformed all the state-of-the-art algorithms regarding all the utilized performance measures. The IAM classifier received 98.08% for accuracy, 98.9% for precision, 98% for recall, and 98.2% for F-measure values. However, the highest results obtained from the other classifiers were 92.87% for accuracy, 92.9% for precision and recall, and 08.8% for F-measure. These results demonstrate the efficiency of IAM for flags prediction.

### 5.6. The Results of the Ionosphere Dataset

The ionosphere dataset has 35 features and 351 records that are classified into two classes. We applied the IAM to this dataset using different splitting parameters to obtain the best performance result. Additionally, we applied the benchmark algorithms using 10-fold cross-validation. The results are presented in Table 8. The highest values are bolded for each performance measure.

The above results of applying IAM and the state-of-the-art classifiers to the ionosphere dataset were relatively low. There are several unknown values for precision, recall, and F-measure due to the division by zero not being computed. However, regarding the accuracy measure, the IAM classifier had the highest value (64.58%), whereas the other classifier’s accuracy values were in the range of 35–62%. These results show that IAM outperformed the other classifiers for ionosphere prediction.

### 5.7. Discussion

To summarize, we proposed the IAM prediction model to avoid running the classifier repeatedly on the entire dataset as it evolves. The model ran on six datasets having a different number of records, features, and classes. All the features of the breast cancer (Ljubljana) dataset are nominal, whereas all the features of the ionosphere and breast cancer (Wisconsin) datasets are numeric. The other datasets, i.e., horse colic, credit card, and flags, have both nominal and numeric features. The number of records in these datasets is in the range from 194 to 699 records. Additionally, these datasets are all binary, as they have two classes, except flags, which has eight different classes. The performance results from running the IAM algorithm and ten state-of-the-art benchmark algorithms on these datasets demonstrated that the IAM classifier outperformed all of the benchmark classifiers, except the bagging and random forest classifiers, where they approximately achieved comparable accuracy for the horse colic dataset. The achieved high accuracy of the proposed model can be attributed to the fact that previously mined classification rules by IAM are fed back to the system, with the newly arrived dataset increment, for pruning and enhancements, which results in an optimized rule set. It is important here to stress that, as IAM runs only on newly arriving increments, it grants better time- and resource-saving. These results demonstrated the strong potential and efficiency of the IAM classifier for prediction objectives in various areas.

## 6. Conclusions

In this paper, we propose IAM, an incremental classifier based on one of the most effective rule mining algorithms, Ant-Miner, to ensure an optimal number of classification rules. IAM dynamically generates optimized classification rules for an evolving dataset using association rule mining and the Ant-Miner algorithm by improving the existing knowledge base on the arrival of new data records. IAM was applied to six datasets having varying number of features, records, and classes. We considered four performance measures: accuracy, precision, recall, and F-measure. To evaluate the performance of the proposed IAM, we compared its performance with 10 rival big data analytics algorithms, namely, naive Bayes, logistic regression, K*, Ada-Boost, bagging, PART, C4.5, multilayer perceptron, support vector machine, and random forest, all of which have previously demonstrated their effectiveness in different areas. These algorithms were run using a 10-fold cross-validation method. According to the experimental results, IAM achieved superior results, which demonstrates its effectiveness. The time efficiency of IAM is undeniable given that it does not require training on the entire dataset as it evolves.

For future work, we aim to improve the proposed IAM classifier and enhance its performance in multi-class datasets. Additionally, the classification rules extracted from the Ant-Miner can be used with other classifiers, such as C4.5, to implement hybridization, which can provide better results. Furthermore, IAM parameters were experimentally set, and the system may benefit more from a self-adapted parameter setting scheme. In additions, the mini batch approach should be tested where each dataset is partitioned into a larger number of increments of smaller sizes.

## Figures and Tables

**Figure 1 sensors-22-02223-f001:**
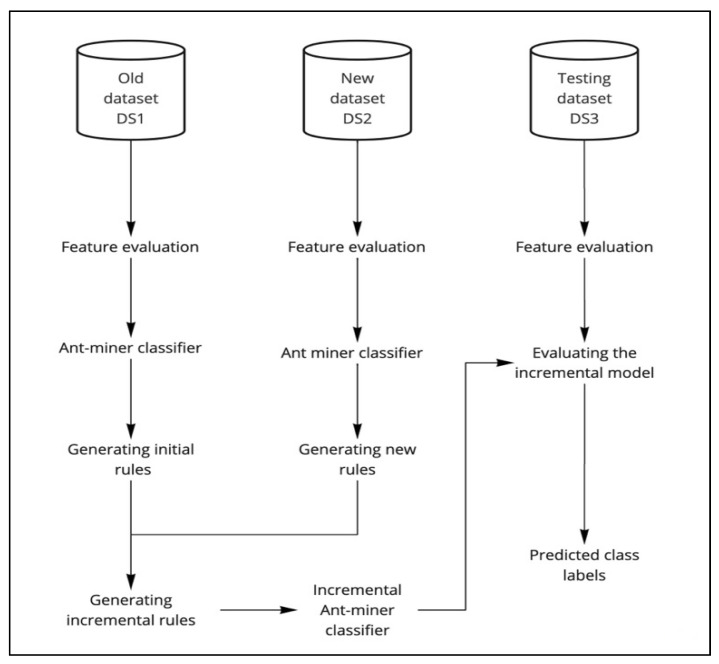
Proposed IAM prediction model.

**Table 1 sensors-22-02223-t001:** List of IAM parameters and their settings.

Parameter	Value	Foundation
Colony size	120	Empirically
Max iterations	3000	Empirically
Minimum cases	10	Default value in Myra
Rule pruner method	backtrack	Default value in Myra
Uncovered examples	10	Default value in Myra
No. of convergence test iterations	10	Default value in Myra
weight value	0.5	Empirically

**Table 2 sensors-22-02223-t002:** Summary of selected datasets.

Dataset Name	No. Records	No. Feature	No. Classes
Breast cancer (Ljubljana)	286	9	2
Breast cancer (Wisconsin)	699	10	2
Horse colic	368	22	2
Credit card	690	15	2
Flags	194	30	8
Ionosphere	351	35	2

**Table 3 sensors-22-02223-t003:** Classifiers’ performances on the breast cancer (Ljubljana) dataset. (The bold numbers are the highest value of each measure).

Algorithm	Accuracy	Precision	Recall	F-Measure
Naïve Bayes	71.68%	77.80%	83.60%	80.60%
Logistic	68.88%	75.20%	83.10%	79.00%
MLP	64.68%	75.00%	74.60%	74.80%
SVM	69.58%	75.00%	85.10%	79.70%
KStar	73.43%	76.80%	89.10%	82.50%
AdaBoost	70.28%	77.10%	82.10%	79.50%
Bagging	69.23%	72.20%	91.50%	80.70%
PART	71.33%	74.50%	90.00%	81.50%
C4.5	75.52%	75.70%	**96.00%**	84.60%
Random forest	69.58%	74.20%	87.10%	80.10%
IAM	**78.18%**	**89.47%**	80.95%	**85.00%**

**Table 4 sensors-22-02223-t004:** Classifiers’ performances on the breast cancer (Wisconsin) dataset. (The bold numbers are the highest value of each measure).

Algorithm	Accuracy	Precision	Recall	F-Measure
Naïve Bayes	95.99%	91.4%	97.5%	94.4%
Logistic	96.56%	95%	95%	95%
MLP	95.85%	93.1%	95%	94%
SVM	96.71%	**94.7%**	95.9%	95.3%
KStar	81.25%	84.8%	55.6%	67.2%
AdaBoost	94.85%	92.9%	92.1%	94.8%
Bagging	95.85%	92.7%	95.4%	94.1%
PART	94.42%	92.4%	91.3%	91.9%
C4.5	94.56%	91.8%	92.5%	92.1%
Random forest	96.85%	94%	97.1%	95.5%
IAM	**98.65%**	94.4%	**100%**	**97.1%**

**Table 5 sensors-22-02223-t005:** Classifiers’ performances on the horse colic dataset. (The bold numbers are the highest value of each measure).

Algorithm	Accuracy	Precision	Recall	F-Measure
Naïve Bayes	77.99%	85.10%	78.90%	81.90%
Logistic	80.99%	86.20%	83.20%	84.60%
MLP	80.43%	85.10%	83.60%	84.30%
SVM	82.61%	85.30%	87.50%	86.40%
KStar	76.63%	83.20%	78.90%	81.00%
AdaBoost	81.25%	84.40%	86.20%	85.30%
Bagging	**86.41%**	86.40%	**93.10%**	89.60%
PART	84.78%	84.90%	92.20%	88.40%
C4.5	85.32%	84.80%	85.30%	88.90%
Random forest	**86.41%**	87.00%	92.20%	89.50%
IAM	**86.11%**	**95.24%**	83.33%	**89.88%**

**Table 6 sensors-22-02223-t006:** Classifiers’ performances on the credit card dataset. (The bold numbers are the highest value of each measure).

Algorithm	Accuracy	Precision	Recall	F-Measure
Naïve Bayes	77.68%	85.90%	59.60%	70.40%
Logistic	85.21%	81.50%	86.30%	83.90%
MLP	83.62%	81.70%	81.40%	81.60%
SVM	84.93%	78.00%	92.20%	84.50%
KStar	78.98%	82.40%	67.10%	74.00%
AdaBoost	84.64%	82.30%	83.40%	82.80%
Bagging	85.65%	81.30%	**87.90%**	84.50%
PART	85.36%	83.20%	84.00%	83.60%
C4.5	86.09%	84.80%	83.70%	84.30%
Random forest	86.96%	85.80%	84.70%	85.20%
IAM	**92.75%**	**93.75%**	86.54%	**90.00%**

**Table 7 sensors-22-02223-t007:** Classifiers’ performances on the flags dataset. (The bold numbers are the highest value of each measure).

Algorithm	Accuracy	Precision	Recall	F-Measure
Naïve Bayes	82.62%	84.2%	82.6%	82.9%
Logistic	88.89%	88.9%	88.9%	88.7%
MLP	91.16%	91.8%	91.2%	90.9%
SVM	88.6%	89.1%	88.6%	88.3%
KStar	84.61%	86.6%	84.6%	83.5%
AdaBoost	90.88%	91.5%	90.9%	90.6%
Bagging	91.16%	91.1%	91.2%	91.1%
PART	91.73%	91.8%	91.7%	91.6%
C4.5	91.45%	91.5%	91.5%	91.3%
Random forest	92.87%	92.9%	92.9%	92.8%
IAM	**98.08%**	**98.9%**	**98%**	**98.2%**

**Table 8 sensors-22-02223-t008:** Classifiers’ performances on the ionosphere dataset. (The bold numbers are the highest value of each measure).

Algorithm	Accuracy	Precision	Recall	F-Measure
Naïve Bayes	40.72%	N/A	40.7%	N/A
Logistic	47.93%	N/A	47.9%	N/A
MLP	60.31%	N/A	60.3%	N/A
SVM	58.76%	N/A	58.8%	N/A
KStar	50%	N/A	50%	N/A
AdaBoost	51.03%	N/A	51%	N/A
Bagging	35.56%	N/A	35.6%	N/A
PART	56.18%	N/A	56.2%	N/A
C4.5	57.73%	N/A	57.7%	N/A
Random forest	62.37%	N/A	**62.4%**	N/A
IAM	**64.58%**	N/A	N/A	N/A

## Data Availability

Not applicable.

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
