# Peer review of "Incremental Ant-Miner Classifier for Online Big Data Analytics"

_sensors, 2022, doi:10.3390/s22062223_

Round 1

Reviewer 1 Report

This manuscript evaluates the Incremental Ant-Miner (IAM) classification algorithm using six different classification datasets. The performance measures of accuracy, precision, recall and f-measure are used and compared with ten rival classifiers. The IAM algorithm was found to be better on most performance measures as well as quicker when new datasets arrive because the classification rules do not have to be re-learned with the entire updated dataset. This is important in the big data analytics domain where new data may arrive in real time.

The Ant-miner algorithm is described in fairly technical terms in Section 3, which I found hard going as unfamiliar with it. If a version more suited to the uninitiated could be given it would make the paper more accessible to all readers.

The paper is well written and the results probably important in many practical machine learning applications. The performance measures used are those appropriate for imbalanced classification datasets, which presumably are exemplified by those chosen for comparison. As algorithm running time is a key criterion (lines 189-190, 525-527), is it possible to give some estimate of what this might be for comparison with off-line machine learning?

A currently popular approach for dealing with incremental data in distributed IoT systems is federated learning, and it would be of interest to compare this with the IAM algorithm (see eg Hu, K., Li, Y., Xia, M., Wu, J., Lu, M., Zhang, S., & Weng, L. (2021). Federated Learning: A Distributed Shared Machine Learning Method. Complexity doi: 10.1155/2021/8261663)

The paper has a tendency to repeat bits of text, eg lines 76-78 versus line 80, lines 420-428 versus lines 510-512 and 534-535.

Reference [40] should be: I. Witten and E. Frank Data Mining: Practical Machine Learning Tools and Techniques, 4th Ed. 2016

Reviewer 2 Report

The contribution looks really interesting but, getting to the examples section, we do not see any specific reason to assume the considered environment to be really dynamic. Specifically, in the reviewer’s opinion the paper can be considered for publication provided that the Authors appropriately account for the following comments:

  1. At the beginning of Section 2, it is repeated a number of times that sub-sections include…part of the survey of the literature. Once said, that looks rather trivial to be repeated over and over.
  2. Sections 2.1 and 2.2 look to be written by two different people: in the first, the name of authors of published papers are always highlighted, whereas in the second they are never mentioned. It is therefore suggested to amend by making the two similar.
  3. At the end of Section 2, it seems that the novelty of this contribution rests on the fact that in the literature online and-miner has never been considered. This should be stressed more, in order not to say that this is the first time but also which benefits it can provide against other algorithms.
  4. Section 3: in the text the existing dataset DS1 is mentioned, but this is not reported explicitly in Fig. 1: text and figure should have the same notation.
  5. Still referring to Fig. 1, see also comments below, the entire dataset is split into a kind of training, validation (though it is not), and testing subsets. This is hard to be really considered as a dynamic environment, wherein data are continuously collected and need to be processed in real-time. Hence, more comments must be added here to understand, again, the benefits of the proposal.
  6. After an Eq., if “where” is used to explain the terms in the equation, it does not have to be capitalized.
  7. After Eq. 1, what is term_ij? Why is it tensorial?
  8. In the explanations to Eqs. 1 and 2, T_ij seems to have two different meanings. Why?
  9. If a variable or a symbol has been already defined, why re-defining them?
  10. Page 6: it is not clear why mentioning Eq. 3 much before the equation itself is provided. If there is a need to introduce it before, then move it up on the same page.
  11. At the end of Section 3.2, it is said what happens if the similarity is =1. But, what happens instead if it is not?
  12. Section 4.2: from this point on, see also comment above, it is not clear why considering an incremental problem in the current setting. Basically, here the dataset is split into two subsets and nothing more. This is not really a dynamic situation. Please, explain a really dynamic one, assuming that a continuous flow of information needs to be processed in real-time.
  13. Section 5: results have been obtained with a rather low-power computer, which is not typically the case when machine learning algorithms are run. Please, explain why? How does it affect the results provided next? The computing time is in fact never provided, and this is an issue if one faces the problem of real-time classification.
  14. Section 5: how are datasets split into subsets? Which target to optimize the split? Why isn’t the classification time never provided? Can the datasets split according to a kind of mini-batch approach to data processing?
  15. In Section 5 the only results commented are those listed already in the Tables. This looks a bit trivial to be repeated every time. Why not add more?
  16. Finally, and we return again the most critical issue of the contribution, in Section 5.7 it is mentioned that the classifier needs to be run repeatedly as “datasets evolve”. But, in this contribution, we do not see any problem characterized by evolving datasets. Results must be added not only to split the datasets in a more “dynamic” situation but also to consider classification results to evolve in time: if results are not expected to change, we do not see any need to allow for more complex algorithms though they can provide (slightly) better results.

Reviewer 3 Report

You included different methods to be compared with yours (IAM). There is no demonstration of the IAM case from A to Z.  The paper is at the edge of full rejection. Use Serbia not Yugoslavia.

Reviewer 4 Report

  1. The manuscript is concerned with incremental ant-miner classifier for online big data analytics, which is interesting. It is relevant and within the scope of the journal.
  2. However, the manuscript, in its present form, contains several weaknesses. Adequate revisions to the following points should be undertaken in order to justify recommendation for publication.
  3. Full names should be shown for all abbreviations in their first occurrence in texts. For example, CN2 in p.2, SVM in p.3, MYRA in p.3, RPF in p.3, etc.
  4. For readers to quickly catch the contribution in this work, it would be better to highlight major difficulties and challenges, and your original achievements to overcome them, in a clearer way in abstract and introduction.
  5. 1 - the Incremental Ant-Miner is adopted for online prediction. What are the other feasible alternatives? What are the advantages of adopting this soft computing technique over others in this case? How will this affect the results? More details should be furnished.
  6. 5 - classifier design as shown in Figure 1 is adopted in this study. What are other feasible alternatives? What are the advantages of adopting this design over others in this case? How will this affect the results? The authors should provide more details on this.
  7. 7 - Equations (3) and (4) are adopted for information-theoretic heuristic function and the entropy. What are the other feasible alternatives? What are the advantages of adopting these equations over others in this case? How will this affect the results? More details should be furnished.
  8. 8 - six datasets in the University of California, Irvine repository [36] are adopted in the experiments. What are the other feasible alternatives? What are the advantages of adopting these datasets over others in this case? How will this affect the results? More details should be furnished.
  9. 9 - ten machine learning algorithms are adopted as benchmark for comparison. What are the other feasible alternatives? What are the advantages of adopting these methods over others in this case? How will this affect the results? More details should be furnished.
  10. 9 - four performance measures are adopted to evaluate the proposed IAM and the benchmark algorithms. What are the other feasible alternatives? What are the advantages of adopting these evaluation metrics over others in this case? How will this affect the results? More details should be furnished.
  11. 13 - a 10-cross validation method is adopted to run the algorithms. What are other feasible alternatives? What are the advantages of adopting this approach over others in this case? How will this affect the results? The authors should provide more details on this.
  12. The discussion section in the present form is relatively weak and should be strengthened with more details and justifications.
  13. Some key parameters are not mentioned. The rationale on the choice of the particular set of parameters should be explained with more details. Have the authors experimented with other sets of values? What are the sensitivities of these parameters on the results?
  14. Some assumptions are stated in various sections. More justifications should be provided on these assumptions. Evaluation on how they will affect the results should be made.
  15. Moreover, the manuscript could be substantially improved by relying and citing more on recent literatures about real-life applications of soft computing techniques in different fields such as the followings. Discussions about result comparison and/or incorporation of those concepts in your works are encouraged:
  • Banan, A., et al., “Deep learning-based appearance features extraction for automated carp species identification,” Aquacultural Engineering 89: 102053 2020.
  • Shamshirband, S., et al., “A Survey of Deep Learning Techniques: Application in Wind and Solar Energy Resources,” IEEE Access 7 (1): 164650-164666 2019.
  • Fan, Y.J., et al., “Spatiotemporal modeling for nonlinear distributed thermal processes based on KL decomposition, MLP and LSTM network,” IEEE Access 8: 25111-25121 2020.
  1. In the conclusion section, the limitations of this study and suggested improvements of this work should be highlighted.

Round 2

Reviewer 2 Report

I appreciate the effort in improving the manuscript.

Anyhow, I still have this concern in reply to the authors' comment "Since, to the best of our knowledge, none of the accessible
datasets evolves over time, we have had to divide available static datasets to mimic the dynamic
arrival of data".

Why the use of algorithms designed for evolving conditions in a static one? The comment here above looks like this: we do not have a dynamic environment, hence we use the method in a static one. Like saying that something done for people on the move is used for people sitting: is that worth it? 

I suggest the authors either find a dynamic condition to test their method or modify the database to introduce a time/instance variation.

Author Response

We really appreciate the time and effort that the reviewer has dedicated to providing insightful comments in our manuscript. We agree that the provided environment is not really a dynamic one. However,  it can be used and is usually employed to model it. But it seems like we have not been clear enough in representing this aspect.  In the following we list the reasons behind dividing available datasets into increments:

  • This methodology of dividing each dataset into three parts is usually followed in similar works for evaluating incremental classifiers, e.g. [11].
  • If we select a dataset that evolves overtime, to apply any incremental classifier, the dataset should be treated in batches (increments). The classifier will not run immediately on the arrival of each single data item. The process works by having the classifier periodically run after the incoming data reach a certain size. In other words, the classifier needs to wait for the incoming data to accumulate, then it runs on the new batch. It is not efficient to run the classifier on the arrival of each single data item.
  • As the name “incremental classifier” suggests, the classifier works incrementally rather than “on the fly”, accordingly, each dataset is divided into increments to realize this concept. 

  With regards to moving and stationary people. We consider the reviewer’s analogy to be apt. But for the sake of clarity, it is perhaps more precise to state that we are capturing the displacement of a moving person at several points in time, and disregarding the velocity as it is not a factor in our model. We hope this helps to clarify our followed approach. Thank you again for the comment.

Reviewer 3 Report

Please, accept the paper in its present form.

Author Response

Thank you so much for reviewing our manuscript and accepting it in its present form.

Reviewer 4 Report

The revised paper has addressed all my previous comments, and I suggest to ACCEPT the paper as it is now.

Author Response

(The authors gave the same response as above.)
